# Protective Effects of Fermented Paprika (*Capsicum annuum* L.) on Sodium Iodate-Induced Retinal Damage

**DOI:** 10.3390/nu13010025

**Published:** 2020-12-23

**Authors:** Ha-Rim Kim, Sol Kim, Sang-Wang Lee, Hong-Sig Sin, Seon-Young Kim

**Affiliations:** 1Jeonju AgroBio-Materials Institute, Wonjangdong-gil 111-27, Deokjin-gu, Jeonju-si, Jeollabuk-do 54810, Korea; poshrim@jami.re.kr (H.-R.K.); sol0819@jami.re.kr (S.K.); 2Chebigen Co., Ltd., Jeonju 54853, Korea; molegene74@hanmail.net (S.-W.L.); shsdo@hanmail.net (H.-S.S.)

**Keywords:** age-related macular degeneration, oxidative stress, *Capsicum annuum* L. (paprika), fermentation, antiiflammation

## Abstract

Diseases of the outer retina, including age-related macular degeneration (AMD), are major cause of permanent visual damage. The pathogenesis of AMD involves oxidative stress and damage of the retinal pigment epithelium. *Capsicum annuum* L. (paprika) fruits have been known as a source of vitamins, carotenoids, phenolic compounds, and metabolites with a well-known antioxidant activity, which have positive effects on human health and protection against AMD and cataracts. In this study, we investigated whether paprika (fermented (FP), yellow, and orange colored) fermented with *Lactobacillus* (L.) *plantarum* could increase the protective effect of retinal degeneration using in vitro and in vivo models. FP significantly increased cell survival and reduced levels of lactate dehydrogenase as well as intracellular reactive oxygen species (ROS) increase in SI (sodium iodate, NaIO_3_)-treated human retinal pigment epithelial (ARPE-19) cells. We developed a model of retinal damage in C57BL/6 mice using SI (30 mg/kg) via intraperitoneal injection. Seven days after SI administration, deformation and a decrease in thickness were observed in the outer nuclear layer, but improved by FP treatment. FP administration protected the SI-mediated reduction of superoxide dismutase and glutathione levels in the serum and ocular tissues of mice. The overproduction of cleaved poly(ADP-Ribose) Polymerase (PARP)1, caspase-3 and -8 proteins were significantly protected by FP in SI-treated cells and ocular tissues. In addition, we evaluated the potentiating effects of FP on antioxidants and their underlying mechanisms in RAW 264.7 cells. Lipopolysaccharide (LPS)-induced nitrite increase was markedly blocked by FP treatment in RAW 264.7 cells. Furthermore, FP reduced LPS-induced inducible nitric oxide synthase and cyclooxygenase-2 activation. The FP also enhanced the inhibitory effects on mitogen activated kinase signaling protein activation in ARPE-19 and RAW 264.7 cells and ocular tissues. There was no significant difference in total phenol and flavonoid content in paprika by fermentation, but the vitamin C content was increased in orange colored paprika, and protective effect against oxidative stress-mediated retinal damage was enhanced after fermentation. These results suggest that FP may be a potential candidate to protect against retinal degenerative diseases through the regulation of oxidative stress.

## 1. Introduction

Age-related macular degeneration (AMD) is a progressive eye disease caused by the degeneration of photoreceptor and retinal pigment epithelial (RPE) cells adjacent in the central part of the macula [1].

Oxidative stress is one of the most important causes of aging process. Studies indicate that oxidative stress contributes to the pathophysiology of AMD. The retina is the most oxygen-consuming tissue, indicating that RPE cells are more susceptible to oxidative stress, especially damage caused by reactive oxygen species (ROS) [2,3,4]. ROS are generated from aerobic metabolism under pathophysiologic conditions that can induce oxidative damage. Imbalanced intracellular ROS can alter essential physiological signaling, such as regulatory feedback related to metabolism, hypoxia, and inflammation, that can impair RPE cell function [5,6].

Cells exposed to stress initially induce activation of compensatory pathways that restore homeostasis; however, chronic or intense stress activates death pathways such as cell death and necrosis [7]. Caspase has an important role in regulating cell apoptosis, and induction of the transformed apoptotic signal cascade and programed cell death. It has been reported that caspase-dependent apoptosis of RPE can be induced by oxidative stress, and inhibition of oxidative stress can effectively protect RPE’s apoptosis. The SI (sodium iodate, NaIO_3_)-induced AMD animal model is widely used to study the molecular mechanism of oxidative stress associated cell death in AMD and induces consistent and selective damage to the RPE. PI3K/AKT has been implicated in a various cellular signaling pathway, including cell survival, differentiation, and death. It has been reported that PI3K/AKT can be involved in SI-induced AMD pathophysiology [8,9,10]. The mitogen-activated protein kinases (MAPKs) are also known to regulate survival and apoptosis [11]. Interestingly, studies indicated that oxidative stress induces RPE cell death and ocular damage via AKT and MAPKs protein kinase cascade signaling [9,10].

ROS induces an inflammatory response that is the expression of cyclooxygenase-2 (COX-2) and inducible nitric oxide synthase (iNOS) [11].

Paprika (*Capsicum annuum* L.) is considered a healthy vegetable that contains phytochemicals including carotenoids, ascorbic acid, tocopherols, flavonoids and phenolic compounds, which are important antioxidant components that may reduce the risk of diseases [12,13,14]. Studies have revealed that paprika displays pharmacological effects such as antioxidant, antimicrobial, anti-inflammatory, anticancer, and antiviral effects [14].

Fermentation is a technology that improves the activity of raw materials through biochemical changes. This bioprocessing is increasingly used in the pharmaceutical, chemical, and food industries as these altered properties appear to produce beneficial effects such as increased antioxidant activity [15,16]. It has been reported that during fermentation, biological activity is enhanced due to the release of flavonoids and phenols, respectively, increased by microbial enzymes in plant chemicals [16].

Despite many reports on useful pharmacological properties of paprika, no studies have been considered using fermented paprika (FP) to improve ocular degeneration and inflammation caused by oxidative stress. In this study, we investigated whether fermented paprika have protective effects on oxidative stress and retinal degeneration. We also tested the expression of proteins involved in apoptosis and MAPK signaling in cells and ocular tissues.

## 2. Materials and Methods

### 2.1. Chemicals and Antibodies

Chemicals and cell culture materials were obtained from the following sources: LPS (lipopolysaccharide), SI (sodium iodate), MTT [3-(4,5)-dimethylthiazolyl-2-yl-2,5-diphenyl tetrazolium bromide], BSA (bovine serum albumin), Folin Ciocalteu reagent, quercetin, gallic acid, and vitamin C (L-ascorbic acid) were purchased from Sigma-Aldrich (St Louis, MO, USA); PBS (phosphate-buffered saline), DMEM (Dulbecco’s modified Eagle’s medium), DME/F12 medium, FBS (fetal bovine serum), penicillin–streptomycin solution, DCF-DA (2′,7′-dichlorofluoresin diacetate) were from Invitrogen (Carlsbad, CA, USA); antibodies against COX-2 and β-actin were from Santa Cruz Biotechnology, Inc.; the protein assay kit was from Thermo Scientific (Waltham, MA, USA); primary antibodies [anti-phospho-ERKl/2 (Thr202/Tyr204), anti-phospho-p38 (Thr180/Tyr182), and anti-phospho-SAPK/JNK (Thr183/Tyr185), anti-AKT/-phospho-AKT (Ser473), anti-(cleaved-)caspase3/8, anti-(cleaved-)PARP1, and secondary antibody (HRP-linked anti-rabbit and anti-mouse IgG) were from Cell Signaling Technology Inc. (Beverly, MA, USA); ECL chemiluminescence system and PVDF (polyvinylidene difluoride) membrane were from Amersham Pharmacia Biotech (Piscataway, NJ, USA). All other chemicals were of analytical grade or complied with the standards required for cell culture experiments.

### 2.2. Fermentation and Preparation of Paprika Samples

*L. plantarum* JBMI F5 (KACC91638P) was prepared following a previously described method [6]. The raw materials thus prepared were mixed using a crusher. The crushed yellow or orange colored paprika (P-Y or P-O) was filtered using a mesh. The prepared paprika solution was adjusted to pH 7.0 after acidification (pH 4.5) for fermentation, sterilization and then inoculated with 5% *L. plantarum* and cultured overnight. The fermentation process was carried out for 18 h while stirring at 150 rpm, 37 °C. After the fermentation, the fermentation broth was recovered and added with dextrin, followed by lyophilization. Raw paprika (-Y or -O), which was inoculated with *L. plantarum* but not fermented, was compared with fermented paprika. The raw or fermented paprika powder (1 mg) was treated with 70% ethanol (10 mL) and sonicated several times at room temperature for 2 h to produce extract. The solvent was evaporated under N_2_ gas atmosphere and freeze-dried. The lyophilized powder and extract was stored at −80 °C for in vivo and in vitro studies, respectively.

### 2.3. Analysis Physicochemical and Antioxidant Effect

Total polyphenol and flavonoids and vitamin C content was analyzed using a previously described method [6,17]. DPPH (2,2-diphenyl-1-picrylhydrazyl) and ABTS (2,2′-azino-bis-3-ethylbezthiazoline-6-sulphonic acid) radical scavenging activity was determined as previously described [6,17].

### 2.4. Cell Culture

Mouse macrophages (RAW 264.7), and human RPE cells (ARPE-19) were obtained from ATCC (American Type Culture Collection). Cells were cultured in DMEM or DMEM/F12 medium (Invitrogen): supplemented with 10% FBS, 100 U/mL penicillin, and 100 μg/mL streptomycin at 37 °C with 5% CO_2_ in a humidified atmosphere.

### 2.5. Cell Viability Assay

The ARPE-19 cells were divided into six groups: Normal (N), SI (sodium iodate), P-Y, FP-Y, P-O, and FP-O pretreated group. ARPE-19 cells were plated at a density of 1 × 10^4^ cells/well in 96-well plates. After cell attachment, the cells were pre-treated with P-Y, FP-Y, P-O, and FP-O for 1 h before SI (1200 μg/mL) treating. After another 24 h, 10 μL of MTT (5 mg/mL) reagent was added to each well, and incubated for 3 h at 37 °C. Then the cell culture medium was removed, and 100 μL of DMSO (dimethyl sulfoxide) was added to each well. The lactate dehydrogenase (LDH) leakage into the culture medium as determined using a commercially available kit from Sigma-Aldrich (St Louis, MO, USA). The absorbance was measured on a microplate reader (Multiskan GO, Thermo Scientific, Waltham, MA, USA) at 570 nm for MTT and 340 nm for LDH assay.

### 2.6. Measurement of Intracellular Reactive Oxygen Species (ROS) Increase

ARPE-19 cells were pretreated with SI for 24 h, followed with or without 24 h exposure to P-Y, FP-Y, P-O, and FP-O extract (500 μg/mL). The intracellular ROS levels in ARPE-19 cells were detected using DCF-DA (Invitrogen. Briefly, cells were incubated with 50 μM DCFH-DA reagent from the kit for 30 min in the dark at 37 °C, then cells were washed twice with PBS. The intracellular ROS was analyzed using a fluorescence microscope (BX53, Olympus, Tokyo, Japan).

### 2.7. Measurement of NO Production in RAW 264.7 Cells

Nitrite, which accumulated in culture supernatants, was measured using a method based on the Griess reaction as an indicator of NO production [18]. Cells (5 × 10^5^ cells/well) were incubated in 6-well plates, and treated with P-Y, FP-Y, P-O, and FP-O extract (500 μg/mL), and then stimulated with LPS (1 μg/mL) for 18 h. The culture medium was mixed with Griess reagent (equal volumes of 1% sulfanilamide in 5% phosphoric acid and 0.1% N-(1-naphtyl) ethylenediamine dihydrochloride), and further incubated for 20 min at room temperature (RT). The concentration of nitrite was determined using the sodium nitrite (NaNO_2_) standard curve at 540 nm with a microplate reader.

### 2.8. Animals and Treatment

C57BL/6J male mice weighing 20–25 g (Damul Science, Deajeon, Korea) were housed in cages with free access to food and water and maintained in temperature and light controlled rooms (23 ± 2 °C, 55 ± 10%, 12/12 h light/dark cycle with lights on at 8:00) at least one week before the experiments. The in vivo experiment was performed in accordance with the Jeonju AgroBio-Materials Institute (JAMI) guidelines under a protocol approved by the Institutional Animal Care and Use Committee (IACUC) of institute and all experiments strictly followed committee guidelines (JAMI IACUC 2019001). Mice were randomly divided into the following six groups (n = 6 per group), which were N (normal control), C (control), P-Y (195 mg/kg), FP-Y (1 × 10^8^ CFU for *L. plantarum* and 195 mg/kg for P-Y), P-O (195 mg/kg), or FP-O (1 × 10^8^ CFU for *L. plantarum* and 195 mg/kg for P-O) treatment group. The raw or fermented paprika powder was suspended in saline. Each of the samples were administered orally once a day for 1 week following SI (sodium iodate, Sigma-Aldrich) treatment. SI was dissolved in sterile normal saline and treated for 5 days by intraperitoneal injection. C group were administered with the normal saline instead of samples. At the end of the study, blood and tissues were harvested from the sacrificed mice after being anesthetized with tribromoethanol (Avertin, Sigma-Aldrich).

### 2.9. Determination of Antioxidant Activity

Superoxide dismutase (SOD) activities and glutathione (GSH) levels were measured analyzed using a SOD (Sigma-Aldrich, St Louis, MO, USA) and GSH (Cayman, Ann Arbor, MI, USA) assay kit, according to the manufacturer’s instructions.

### 2.10. Histology of Ocular Tissues

Ocular tissues were removed and immediately placed in 10% formalin solution, embedded in paraffin and cut into 5-μm sections. Specimens were stained with hematoxylin and eosin (H and E) to identify morphological changes. Histological images were acquired by light microscopy (Nikon, Tokyo, Japan).

### 2.11. Immunoblotting

RAW 264.7 and ARPE-19 cells and ocular tissues were lysed in ice-cold lysis buffer [10 mM Tris-HCl (pH 7.4), 0.1 M ethylenediaminetetra acetic acid (EDTA), 10 mM NaCl, and 0.5% Triton X-100] supplemented with a protease and a phosphatase inhibitor cocktail (Sigma-Aldrich). Protein concentrations were determined by protein assay (ThermoFisher Scientific) using BSA (Sigma-Aldrich) as a standard. Lysates (20 μg) were resolved on a 10–12% sodium dodecyl sulfate-polyacrylamide gel electrophoresis (SDS-PAGE) gel and then transferred to PVDF membranes (BioRad, Hercules, CA, USA). After blocking with Tris-buffered saline (TBS) containing 5% nonfat dry milk and 0.1% (*w/v*) Tween 20, membranes were probed with primary antibody. The following specific antibodies were used to characterize protein expression: iNOS, COX-2, (p-)AKT, (p-)ERK1/2, (p-)JNK, (p-)p38, (cleaved-)Caspase 3/8, (cleaved-)PARP1, and β-actin. After overnight incubation at 4 °C, then incubated with secondary antibody (anti-rabbit IgG-HRP or anti-mouse IgG-HRP). Immunoblots were visualized using chemiluminescent (ECL) Western blotting detection reagents (GE Healthcare, Buckinghamshire, UK) and immunoreactive signals were analyzed by densitometry scanning (Amersham imager 600, GE Healthcare, Buckinghamshire, UK).

### 2.12. Statistical Analysis

Data represent means ± standard deviation (SD) of at least three separate experiments. Data were analyzed using Sigmaplot v14.0 software (Systat Software Inc., San Jose, CA, USA). Student’s *t*-test was performed to compare the parameters between two groups, while the analysis of variance (ANOVA) test followed by Duncan’s multiple range test. Data with *p* < 0.05 were considered statistically significant.

## 3. Results

### 3.1. Physicochemical Content and Radical Sacavenging Ability

In this study, we tested using yellow and orange paprika (P-Y and P-O) on eye damage protection. The raw and fermented paprika was extracted using 70% ethanol for in vitro study. After fermentation, the physicochemical contents and antioxidant effects was compared with raw paprika. As shown in Table 1, the total phenolic and flavonoid levels and their antioxidant activity in paprika was increased by fermentation. The phenolic compounds usually have glycosidic bonds with various glycosides, the bonds of which were hydrolyzed by microorganisms for the release of the total phenolic and flavonoid [19] content. The phenolic compounds are known to have many potential benefits for human health, especially as strong antioxidants [20,21]. The total phenol content of fermented yellow or orange paprika (FP-Y or FP-O) was measured at the level of 84.17 or 81.53 mg GAE (gallic acid equivalent) g^−1^ and the flavonoid content was 9.59 or 8.69 mg QE (quercetin equivalent) g^−1^. Each level did not increase significantly compared to raw paprika. On the other hand, vitamin C content was increased in FP-O compared to P-O. The antioxidant activity was significantly increased by fermentation compared to raw paprika.

### 3.2. Effects of Fermented Paprika (FP) on Sodium Iodate (SI)-Induced Damage in Human Retinal Pigment Epithelial Cell line (ARPE-19)

The effect of FP on SI-mediated cytotoxicity was determined using dimethylthiazolyl-2-yl)2,5-diphenyl tetrazolium bromide (MTT) and lactate dehydrogenase (LDH) leakage assay. ARPE-19 cells were exposed to SI (1200 μg/mL) for 24 h in presence or absence of various concentrations ranging from 1 to 1000 μg/mL of P (-Y or -O) and FP (-Y or -O) extracts (Figure 1A,B). As shown in Figure 1A,B, SI (1200 μg/mL) significantly decreased the viability of ARPE-19 cells compared with the normal (N) group. Otherwise, SI-induced cell damage was protected by P (-Y or -O) and FP (-Y or -O) extracts in a dose dependent manner. Cell viability was increased significantly at doses of 500 μg/mL FP (-Y or -O) compared to the raw extract. The effects of P and FP extract on SI-induced cell damage was further confirmed using LDH leakage into the cell culture medium. As shown in Figure 1C, P (-Y or -O) and FP (-Y or -O) significantly suppressed SI-induced LDH leakage at a dose of 500 μg/mL. In addition, FP-O showed the most potent effect compared to other groups. The cytotoxicity was not observed in all samples at a dose from 1 to 1000 μg/mL. As shown in Figure 1D, no significant difference was observed among various groups. Therefore, the most effective and non-cytotoxic concentration of P(-Y or -O) and FP (-Y or -O) extracts, 500 μg/mL, was used in subsequent studies.

### 3.3. Effects of FP on SI-Induced Intracellular ROS Generation in ARPE-19 Cells

The mechanism of SI-induced retinal damage is associated with oxidative stress, but the protective effect of FP on the SI-induced ROS generation in RPE cells was not revealed. In order to investigate the effect of FP on oxidative stress in ARPE-19 cells, the SI-induced intracellular ROS generation were determined using DCF-DA. As shown in Figure 2, intracellular ROS was markedly increased in APPE-19 cells by SI treatment, while the P (-Y or -O) and FP (-Y or -O) extract reduced the intracellular ROS increase. Moreover, FP (-Y or -O) extract caused a significant (*p* < 0.05) decrease in the ROS levels, compared to raw materials (Figure 2B). These results suggested that the enhanced protective effect against cell damage by SI of P (-Y or -O) is due to increased antioxidant capacity by fermentation.

### 3.4. Effect of FP on Lipopolysaccharide (LPS)-Induced Nitric Oxide (NO) Production in RAW 264.7 Macrophages

Reactive nitrogen species (RNS) such as nitric oxide (NO) also contribute to the oxidative stress. NO is produced by three different isoforms of nitric oxide synthases (NOs), which catalyze L-arginine to L-citrulline by releasing the NO. Cellular NO causes the production of several RNS that are implicated in oxidative damage [22]. In this regard, RAW 264.7 macrophages were included to further confirm the effect of fermented paprika extract (FPE) on oxidative stress. LPS-induced NO production was suppressed by P (-Y or -O) and FP (-Y or -O) extracts pretreatment at 500 μg/mL (Figure 3A). The inhibitory effect of LPS-induced NO increase was more effective in cells pretreated with FP (-Y or -O) extracts than their raw counterparts (Figure 3A). The expression levels of NO production-related proteins, including iNOS and COX-2, were significantly up-regulated in the LPS-treated group (C, control) by 2.65- and 3.46-fold (*p* < 0.001), compared with the N group (Figure 3B). Raw or fermented paprika extract significantly reduced LPS-stimulated iNOS and COX-2 protein expression levels in RAW 264.7 cells (Figure 3B). Consistent with the results in Figure 2, fermented paprika extracts exhibited a more potent inhibitory effect on LPS-mediated NO production and signaling protein expression in RAW264.7 cells.

### 3.5. Preservation Efect of Fermented Paprika Extract (FPE) on Retinal Structure and Oxidative Stress in SI-Induced Retinal Degeneration Mose Model

Oxidative stress could be involved in the photo-aging process as a primary factor. The above in vitro results suggest that fermented paprika extract may be an effective means of protecting retinal damage from oxidant insult and consequent damage. To validate whether the FP provides protection against SI-induced retinal degeneration, we established a mouse model of SI-induced retinal degeneration. SI is an oxidative toxic agent and widely used because of its reproducibility and controllable retinal damage degree [23,24]. For in vivo studies, lyophilized P (-Y or -O) and FP (-Y or -O) powders were suspended in sterile saline and administered orally once a day for 1 week. The dosage for each sample was 1 × 10^8^ CFU for *L. Plantarum*, 159 mg/kg for paprika, and 500 mg/kg for total sample amount.

Representative images from histologic sections stained with hematoxylin and eosin are presented in Figure 4A. As expected, the outer nuclear layer (ONL) cells of SI-induced retinal degeneration control mice (C) were disordered and thinner, and the structure of the RPE layer was not clear compared to normal mice (N). In contrast, administration of P (-Y or -O) and FP (-Y or -O) rescued RPE degeneration and retinal structure (Figure 4A). Compared to the control (C, SI-treatment group), the FP (-Y or -O) group was effective in preventing ONL thickness reduction and improving shape change (Figure 4A). Because oxidative stress has been identified as a major inducer of RPE injury in a SI-induced retinal degeneration model [25], we further assessed the antioxidant capacity of FP in the serum and tissue on the basis of the superoxide dismutase (SOD) and glutathione (GSH) assay. As shown in Figure 4B,C, SOD and GSH content significantly decreased in serum and tissues of the mice with SI injection (*p* < 0.001), whereas FP (-Y or -O) treatment significantly increases the levels of SOD and GSH (*p* < 0.001) (Figure 4B,C). These data suggest that FP shows increased anti-oxidative effects against SI-induced oxidative stress in mice ocular tissue.

### 3.6. FP Inhibits SI-Induced Apoptosis in ARPE-19 Cells and Mouse Retina

SI treatment reduces vision and follows RPE damage with photoreceptor degeneration [25,26]. We also observed similar results in which SI treatment induces cells death, as measured by LDH release in ARPE-19 cells (Figure 1D). The cell death process of apoptosis is responsible for the cell loss seen in several disorders of retina, glaucoma, and macular degeneration [27,28]. Evidence indicates that caspases play a key role in both the initiation and execution pathways of apoptosis [29,30]. Similarly, the cleavage of PARP-1 by caspases is also considered to be a hallmark of apoptosis [31].

To identify whether the FP could affect SI-induced apoptosis in ARPE-19 cells and in mouse retina, we analyzed cell death-related signaling mechanisms. Indeed, our data show that SI induces proteolytic cleavage and activation of PARP-1 and both caspase-8 and -3 in the ARPE-19 cells and mouse retina (Figure 5). However, no significant decrease was observed in the expression of cleaved-PARP-1 and caspase-8 and -3 by treatment with P (-Y or -O) and FP-Y extract whereas PF-O extract significantly inhibited SI-induced cleavage of PARP-1 and caspase-8 and -3 in ARPE-19 cells (Figure 5A; *p* < 0.001). Similar to the results in cells, FP-O administration showed the most significant effect on the SI-induced apoptosis in mouse retina. Additionally, FP-Y showed an effect on the cleavage of caspase 3 (Figure 5B). These results suggest that FP, especially FP-O, could inhibit SI-induced retinal apoptosis via the PARP-1 and caspases pathways.

### 3.7. Effect of FP on SI-Mediated AKT and Mitogen-Activated Protein Kinases (MAPK) Signaling In Vitro and In Vivo

To further confirm the capacity of FP, the effects on SI-induced oxidative stress-related signaling pathways were investigated in cellular and retina tissue. Oxidative stress is known to be a potent activator of PI3K/AKT and MAPKs [9,32]. The phosphorylation of the signaling molecules, which are AKT, ERK1/2, JNK, and p38, were significantly increased by SI administration in ARPE-19 cells (Figure 6A) and mouse retina (Figure 6B). There is no significant difference between C and P (-Y or -O) and FP-Y extract treatment group in ARPE-19 cells. Interestingly, FP-O extract showed an inhibitory effect on SI-induced p38 phosphorylation in ARPE-19 cells (Figure 6A). Furthermore, FP-O extract markedly decreased the phosphorylation of AKT, JNK, and p38 compared to the C group (Figure 6A). On the other hand, SI-induced ERK1/2 phosphorylation was not affected by FP (-Y or -O) as well as its raw materials extract treatment in ARPE-19 cell. Further analysis in mouse retina showed that the FP (-Y or -O) treatment significantly downregulated the phosphorylated AKT, ERK1/2, JNK, and p38 compared to SI treatment group (Figure 6B), indicating that fermentation of paprika, may be helpful for regulation of SI-induced PI3K/AKT and MAPK activation.

We further examined whether FP affects LPS-induced MAPK phosphorylation in RAW 264.7 macrophages. RAW 264.7 cells were pretreated with P (-Y or -O) and FP-Y extract (500 μg/mL) and then treated with LPS (1 μg/mL) for 18 h. As shown in Figure 7, treatment with LPS caused a significant increase in phospho-ERK1/2, JNK, and p-38 MAPK. FP extract significantly decreased the LPS-stimulated phosphorylation of JNK and p-38, whereas it had no effect on the expression level of p-ERK1/2. In both cellular levels, we obtained similar results that ERK1/2 phosphorylation was not affected by FP treatment.

These results suggest that the ability of fermented paprika to inhibit phosphorylation of JNK and p38 MAPK mediated by LPS in the RAW 264.7 macrophage.

## 4. Discussion

In an attempt to understand the protective role of fermented paprika in oxidative stress-induced RPE degeneration, we have investigated the effect of suppression of FP on apoptosis and have studied the signaling mechanisms associated with this effect. For this purpose, we used a murine model of NaIO_3_-induced retinal degeneration in vivo and in human RPE and LPS-induced oxidative stress in RAW264.7 macrophages. Oxidative stress has long been recognized as a contributing factor to the pathogenesis of AMD [2,4,5]. A commonly used experimental model to study the link between oxidative stress and AMD involves the use of cultured human RPE (ARPE19) cells [4,8,11]. Thus, it would be of great significance to explore the oxidative damage of RPE cells in order to examine the pathogenesis and treatment of AMD [5,24,33].

Studies have reported that *Capsicum annuum* L. exhibit a range of bioactivities including antioxidant, antimicrobial, antiviral, anti-inflammatory, and anticancer properties [12,13,14]. Health-linked functionality improves by fermentation of nutrients, which is useful technology for enhancing antioxidative and biological activity [15,20]. *L. plantarum* is a lactic acid species that improves the content of phenolic compounds and subsequently enhanced antioxidant capacity during fermentation [34]. Recently, fermentation has been reported as one of the effective means to improve the antioxidant properties of plants [6,34]. We also reported that antioxidant effects of blackberry were increased by fermentation using *L. plantarum* JBMI F5 [6]. However, studies on the beneficial effects of fermentation to improve biological efficacy have been reported using various materials, but there is no research on fermented paprika.

Based on this background, we tested whether paprika fermented with *L. plantarum* could increase its antioxidant capacity and the ability to regulate retinal degeneration which caused by oxidative stress.

Total antioxidant capacity after fermentation was compared to non-fermented paprika. The radical scavenging effect of the fermented paprika was increased by fermentation (Table 1). Intracellular oxidative damage and consequent dysfunction in various pathological conditions is caused by increased cellular production of ROS and RNS [22]. ARPE-19 cells and RAW 264.7 cells were used to determine the potential of FP for intracellular oxidative stress. FP significantly improved the effect of reducing intracellular ROS production in ARPE-19 cells and NO production in RAW 264.7 cells (Figure 2 and Figure 3).

SI as a retinotoxin that is directly toxic to RPE by oxidative stress and consequently induces retinal degeneration [2,4,5]. To confirm that paprika fermentation is more useful for treating oxidative stress-induced ocular degeneration, we used a SI-induced retinal degeneration model in ARPE-19 cells and mice. Figure 4A shows that SI treatment induced extensive disruption in the ONL and RPE. In contrast, pretreatment with FP ameliorated SI-induced retinal degeneration by counteracting oxidative stress through decreasing the SOD and GSH levels in serum and tissues (Figure 4B,C). These in vivo experimental results suggested that, FP may protect the retinal oxidative stress through those functions mentioned in this study.

SI has been reported to induce RPE cell death, which is associated with PARP and caspase-dependent cell death [24,26,31]. In order to evaluate if FP has the anti-apoptotic effects on SI-induced apoptosis, we analyzed apoptosis-related signaling proteins using Western blot analysis in APRE-19 cells and mouse retina tissues. The results showed that FPE had an anti-apoptotic effect on reducing the levels of cleaved PARP-1 and caspase-8 and -3 (Figure 5).

Next, we further examined the FP effect on ROS signaling pathway in vitro and in vivo. We found that FP administration inhibits AKT, JNK, and p38 phosphorylation, which are induced by SI treatment in ARPE-19 cells and retinal tissues (Figure 6). ROS modulated various signaling pathways involved in inflammation, proliferation, differentiation, and apoptosis. Apoptosis is regulated by several signaling mechanisms, including the AKT and mitogen activated kinases in mammalian cells. Studies have reported that ROS can activate PI3K/AKT as well as MAPK subfamilies including ERK1/2, JNK, and P38 [9,11]. Hao et al. have reported that piceatannol protects H_2_O_2_-induced ARPE-19 cells’ apoptosis through inhibiting AKT phosphorylation [10]. Accumulating evidence reported that MAPKs are implicated in the pathogenesis of many disease [11,32,35]. In RPE cells, inhibition of MAPK pathway was able to prevent oxidative stress-induced RPE apoptosis [11]. Three conventional MAPKs subfamilies, ERKs, JNK, and p38-MAPK, play an important role in modulating the apoptotic pathway. The MAPKs pathway also plays important role in the transcriptional regulation of the LPS-induced expression of iNOS and COX-2 in macrophages [35].

Studies have shown that phenolic compounds increase during fermentation. However, in our results, TPC and TFC did not change significantly by fermentation (Table 1). These results suggest that fermentation by *L. plantarum* contributed to glycoside cleavage and then depolymerization of additional polymer compounds [34]. However, our hypothesis will have to be proved through further research.

In summary, our in vitro and in vivo results have shown that fermented paprika can increase the ability to protect macrophages and retinal damage caused by oxidative stress. The protective effect of FP on oxidative stress-mediated damage is the result of its anti-apoptotic effect and regulation of PI3K/AKT and MAPK signaling. In conclusion, FP can be expected as a potent therapeutic candidate for the treatment of retinal degenerative diseases.

## Figures and Tables

**Figure 1 nutrients-13-00025-f001:**
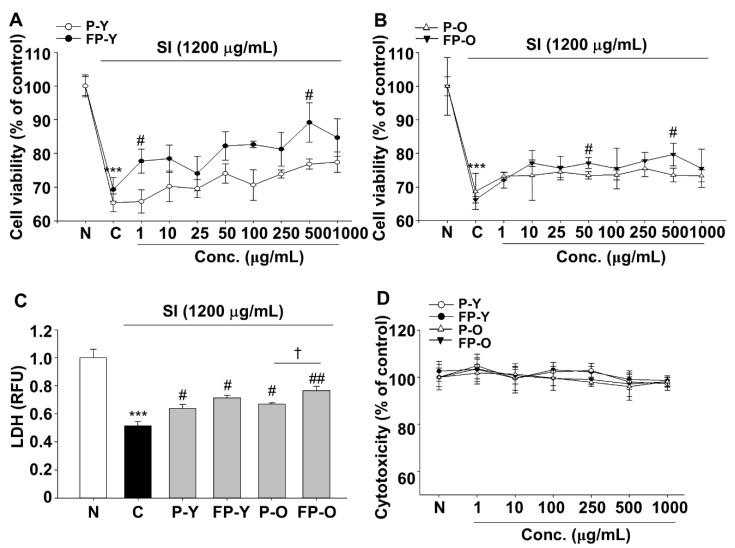
Sodium iodate (SI)-induced damage of human RPE cells (ARPE-19) was decreased by fermented yellow or orange paprika (P-Y or -O) extracts. Cells were exposed to SI (1200 μg/mL) in presence or absence of various doses of P (-Y or -O) and FP (-Y or -O) extracts for 24 h. Protective effects of (**A**) P-Y, FP-Y, (**B**) P-O, and FP-O extracts on SI-induced damage in ARPE-19 cells. (**C**) Cell viability was determined using LDH leakage assay. The SI-induced cell damage was reduced at 500 μg/mL of P (-Y or -O) and FP (-Y or -O) extracts. (**D**) Cell cytotoxicity was analyzed using the MTT assay. *** *p* < 0.001 vs. N, # *p* < 0.05, ## *p* < 0.01 vs. C) † *p* < 0.05 vs. P-O. Values are means ± SEM. Significant differences between the groups were determined by analysis of variance (ANOVA) followed by Duncan’s multiple range test. Conc., Concentration; N, Normal; C, Control; P-Y, Yellow Paprika; P-O, Orange Paprika; FP-Y, Fermented P-Y; FP-O, Fermented P-O.

**Figure 2 nutrients-13-00025-f002:**
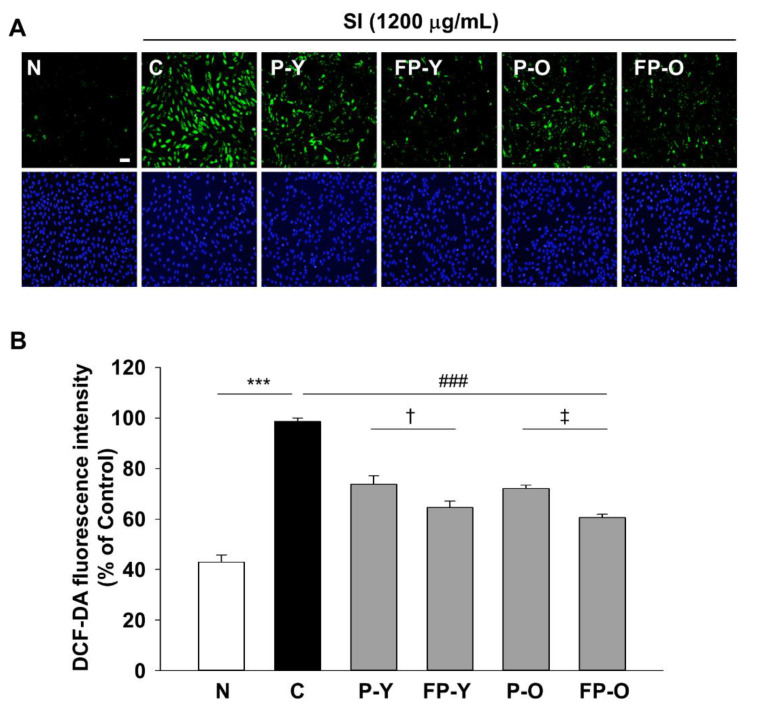
Effects of fermented paprika (FP) on SI-induced ROS generation in ARPE-19 cells. Cells were exposed to SI after incubated with 500 μg/mL of P (-Y or -O) and FP (-Y or -O) extract for 24 h. Intracellular ROS were measured using DCFH-DA (2′,7′-dichlorofluoresin diacetate) assay. The magnification was ×200. Scale bar = 20 μm. (**A**) Representative fluorescence micrographs depicting ROS generation in SI-exposed ARPE-19 cells for 24 h after pretreatment with P (-Y or -O) and FP (-Y or -O) extract at 500 μg/mL (upper panel). Total (DAPI stained) number of cells were presented in lower panel. (**B**) DCF-DA intensity was quantified using Image J. *** *p* < 0.001 vs. N, ### *p* < 0.01 vs. C, † *p* < 0.05 vs. P-Y, ǂ *p* < 0.01 vs. P-O. Values are means ± SEM. Significant differences between the groups were determines by ANOVA followed by Duncan’s multiple range test. N, Normal; C, Control; P-Y, Yellow Paprika; P-O, Orange Paprika; FP-Y, Fermented P-Y; FP-O, Fermented P-O.

**Figure 3 nutrients-13-00025-f003:**
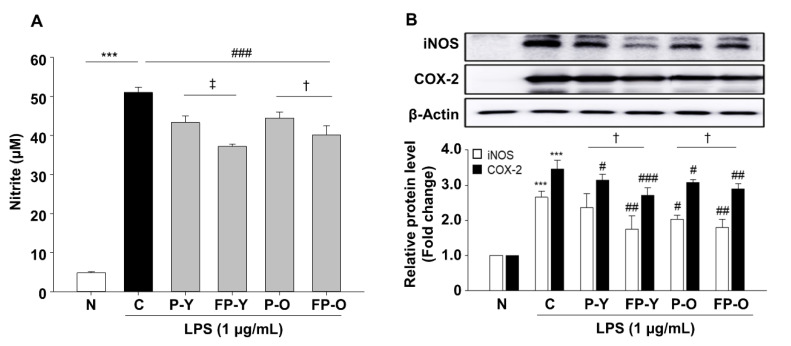
Effects of FP on LPS-induced NO production and iNOS, and COX-2 protein expression in RAW 264.7 cells. RAW264.7 cells were pretreated with P (-Y or -O) and FP (-Y or -O) extracts (500 μg/mL) for 1 h, and then incubated with LPS (1 μg/mL) for 18 h. (**A**) The nitrite levels in culture media were measured using Griess reagent. (**B**) The protein levels of iNOS and COX-2 were analyzed by immunoblotting in cell lysates. The band density was normalized to β-Actin followed by statistical analysis. *** *p* < 0.001 vs. N; # *p* < 0.05, ## *p* < 0.01, ### *p* < 0.001 vs. C. † *p* < 0.05 vs. P (-Y or -O), ǂ *p* < 0.01 vs. P-Y. Values are means ± SEM. Significant differences between the groups were determines by ANOVA followed by Duncan’s multiple range test. LPS, Lipopolysaccharides; iNOS, inducible nitric oxide synthase; COX, cyclooxygenase; N, Normal; C, Control; P-Y, Yellow Paprika; P-O, Orange Paprika; FP-Y, Fermented P-Y; FP-O, Fermented P-O.

**Figure 4 nutrients-13-00025-f004:**
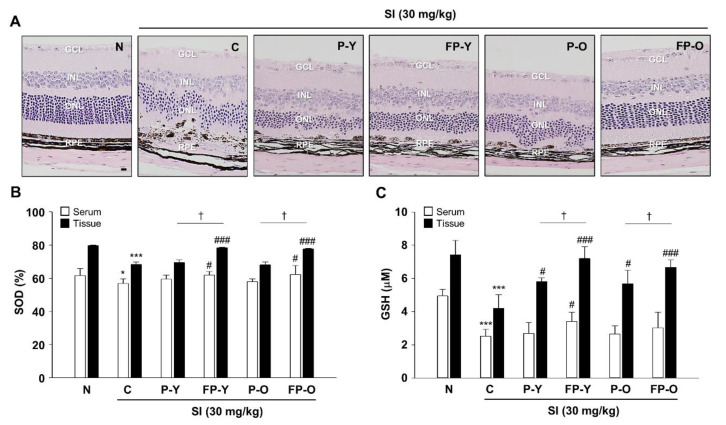
FP protect SI-induced retinal morphological impairment in mice. (**A**) The enucleated eyes were stained with hematoxylin and eosin (H and E). Histological images showing the alterations in retinal morphology. The magnification was ×200. Scale bar = 20 μm. (**B**) Superoxide dismutase (SOD) and (**C**) glutathione (GSH) levels were measured in serum and tissue lysates from all groups using an enzyme-linked immunosorbent assay (ELISA) kit. * *p* < 0.05, *** *p* < 0.001 vs. N; # *p* < 0.05, ### *p* < 0.001 vs. C. † *p* < 0.05 vs. P (-Y or -O). Values are means ± SEM. Significant differences between the groups were determines by ANOVA followed by Duncan’s multiple range test. GCL, Ganglion cell layer; INL, Inner nuclear layer; ONL, Outer nuclear layer; RPE, Retinal pigment epithelium; N, Normal; C, Control; P-Y, Yellow Paprika; P-O, Orange Paprika; FP-Y, Fermented P-Y; FP-O, Fermented P-O.

**Figure 5 nutrients-13-00025-f005:**
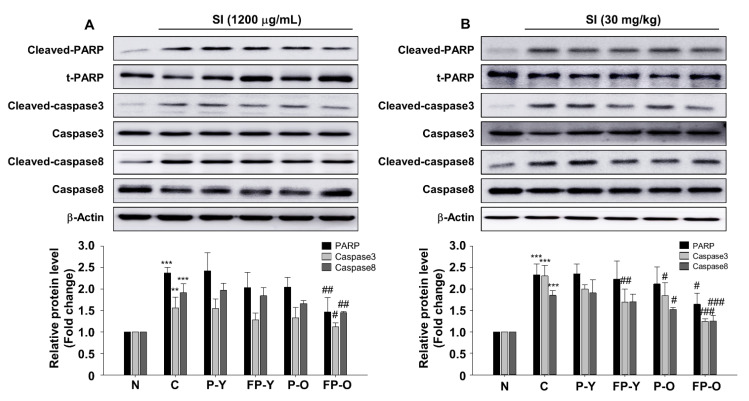
FP inhibits SI-induced apoptosis in APRE-19 cells and mouse retina. Representative blot and quantitative analysis of PARP-1, caspase 3, and caspase 8 protein expression levels in (**A**) ARPE-19 cells and (**B**) mouse retina. Each band was densitometrically quantified by image analysis. The band density was normalized using each total protein. ** *p* < 0.01, *** *p* < 0.001 vs. N; # *p* < 0.05, ## *p* < 0.01, and ### *p* < 0.001 vs. C. Values are means ± SEM. Significant differences between the groups were determines by ANOVA followed by Duncan’s multiple range test. PARP, poly (ADP-ribose) polymerase; N, Normal; C, Control; P-Y, Yellow Paprika; P-O, Orange Paprika; FP-Y, Fermented P-Y; FP-O, Fermented P-O.

**Figure 6 nutrients-13-00025-f006:**
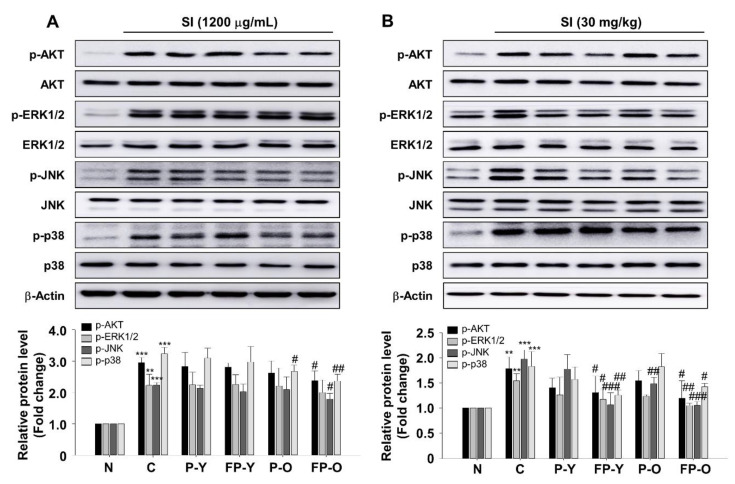
Effect of FP on ultraviolet (UVB)-induced AKT and MAPKs phosphorylation in APRE-19 cells and mouse retina. Representative blot and quantitative analysis of AKT, p-AKT, ERK, p-ERK1/2, JNK, p-JNK, p38, and p-p38 in (**A**) ARPE-19 cells and (**B**) mouse retina. β-Actin was used as a loading control. The band density was normalized using each total protein. Each band was densitometrically quantified by image analysis. ** *p* < 0.01, *** *p* < 0.001 vs. N; # *p* < 0.05, ## *p* < 0.01, and ### *p* < 0.001 vs. C. Values are means ± SEM. Significant differences between the groups were determined by ANOVA followed by Duncan’s multiple range test. N, Normal; C, Control; P-Y, Yellow Paprika; P-O, Orange Paprika; FP-Y, Fermented P-Y; FP-O, Fermented P-O.

**Figure 7 nutrients-13-00025-f007:**
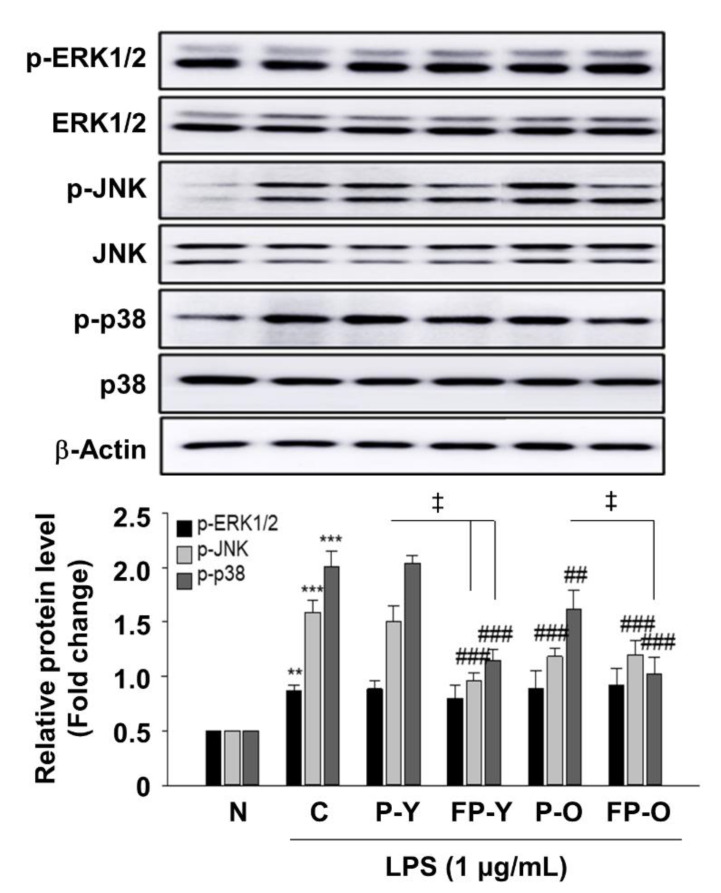
Effect of FP on LPS-induced MAPKs activation in RAW 264.7 macrophages. Levels of phosphorylated ERK1/2, JNK, and p38 were assessed by immunoblot analysis. β-Actin was used as a loading control. Each band was densitometrically quantified by image analysis. ** *p* < 0.01, *** *p* < 0.001 vs. N; ## *p* < 0.01, and ### *p* < 0.001 vs. C; ǂ *p* < 0.01 vs. P-Y or P-O. Values are means ± SEM. Significant differences between the groups were determines by ANOVA followed by Duncan’s multiple range test. LPS, Lipopolysaccharides; N, Normal; C, Control; P-Y, Yellow Paprika; P-O, Orange Paprika; FP-Y, Fermented P-Y; FP-O, Fermented P-O.

**Table 1 nutrients-13-00025-t001:** Physicochemical content and radical scavenging ability.

	TPC (GAE, mg/g)	TFC (QE, mg/g)	VC (mg/100 g)	DPPH Radical Scavenging Activity (%)	ABTS Radical Scavenging Activity (%)
P-Y	81.59 ± 0.01	8.78 ± 0.04	1.98 ± 0.02	24.29 ± 0.56	48.25 ± 1.21
P-O	70.52 ± 0.11	8.00 ± 0.09	1.88 ± 0.02	25.09 ± 0.86	50.70 ± 0.39
FP-Y	84.17 ± 0.54	9.59 ± 0.85	2.05 ± 0.07	34.51 ** ± 0.54	55.14 ** ± 1.28
FP-O	81.53 ± 0.40	8.69 ± 0.29	2.03 ^#^ ± 0.03	28.26 ^##^ ± 1.22	53.45 ^#^ ± 0.24
AA	-		-	95.52 ± 0.12	-
Trolox	-		-	-	94.66 ± 0.06

Values are means ± SD of three independent experiments. ** *p* < 0.01 vs. P-Y; ^#^
*p* < 0.05, ^##^
*p* < 0.01 vs. P-O. TPC, total polyphenol content; GAE, gallic acid equivalent; TFC, total flavonoid content; QE, quercetin equivalent; VC, vitamin C content; DPPH; 2,2-diphenyl-1-pycrylhydrazyl; ABTS, 2,2′-azino-bis (3-ethylbenzothiazoline-6-sulphonic acid); P-Y, Yellow Paprika; P-O, Orange Paprika; FP-Y, Fermented P-Y; FP-O, Fermented P-O; AA, Ascorbic acid.

## Data Availability

Data available on request due to ethical restrictions.

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
