# Peer review of "Protective Effects of Fermented Paprika (Capsicum annuum L.) on Sodium Iodate-Induced Retinal Damage"

_nutrients, 2020, doi:10.3390/nu13010025_

Round 1
Reviewer 1 Report
This manuscript evaluated the effect of Fermented Paprika Extract (EPR) on sodium iodate induced Retinal damage. The authors test the EPR effect on ARPE-19 cells, RAW 264.7 Macrophages, and in vivo mouse retina. There are some major concerns:
- This work does not appear to be major advance beyond the reference20, 21.
- Several models used in this research. Why not keep in one cell model?
- In table 1 ABTS radical scavenging activity, there is no significant difference between YP and FYP?
- Please show N in each group in all figures and table.
- Based figure 1A, it seems YP is better than FYP.
Author Response
Response to the Reviewer’s Comments (Reviewer 1)
Manuscript number: Nutrients-1003076
Title: Protective Effects of Fermented Paprika Extract (Capsicum annuum L.) on Sodium Iodate-induced Retinal Damage
We appreciate very much reviewers’ kind and helpful assessment of our manuscript. We also thank the reviewer for the effort and time put into the review of the manuscript. We have carefully considered and addressed reviewers’ comments, point by point. The changes in the revised manuscript have been marked with colored letters.
Responses on the specific comments and changes in the revised manuscript are as follows:
This manuscript evaluated the effect of Fermented Paprika Extract (EPR) on sodium iodate induced Retinal damage. The authors test the EPR effect on ARPE-19 cells, RAW 264.7 Macrophages, and in vivo mouse retina. There are some major concerns:
- This work does not appear to be major advance beyond the reference20, 21.
This study may be useful in the treatment of oxidative stress-induced eye damage that is enhanced by fermentation. It is important that no studies have yet been reported on this using fermented paprika.
- Several models used in this research. Why not keep in one cell model?
The reason, why we showed the effect of fermented paprika using various cell line, is that the major purpose of this study is focused on enhanced antioxidant and protective effect on oxidative stress mediated pathogenesis of paprika after fermentation.
- In table 1 ABTS radical scavenging activity, there is no significant difference between YP and FYP?
We rechecked the significant difference all parameters including YP and FYP. The corrected results were presented and highlighted in table 1.
- Please show N in each group in all figures and table.
Following your kind advice, we showed N in all figures
- Based figure 1A, it seems YP is better than FYP.
As you mentioned, the YP has been changed into FYP, there is a mistypo.
Furthermore, we modified the abbreviations entirely to reduce confusion.
Best Regards,
Submission Date
31 Oct 2020
Date of this review
30 Nov 2020

Reviewer 2 Report
In their article the authors found that antioxidant properties of fermented paprika may be interesting to prevent oxidative stress cellular damage in RPE. Although the results are interesting there are some methodological concerns that require clarification.
Minor points:
- The meaning of the abbreviations FYP and PFO should appear in the main text. I only could find those meanings in a table legend.
- Page 2 lines 54 to 60. The introduction should include the objectives and the antecedents of the research, I suggest to revise the redaction of those paragraphs.
- Page 3 line 109. HEK293 cells were not used in any of the experiments.
- Page 9 line 301. The authors should revise the sentence since they did not measure vision
Major points:
- Statistical analysis. There are six groups. They should be compared together and not in groups of two or three
- Although there are six animal per group, the results are expressed as the mean of three separate experiments. Is that enough for an ex vivo treatment?
- Western blot results: How was calculated the “relative protein level” shown in bar graphs? Especially in those from those obtained from tissues from animals under the different treatments. It seemed to me that they quantified the fold increase comparing to “N” group”. Is the result the same when comparing the normalized densitometry of each group? There are some differences between the bars and the representative blots that should be carefully revised.
- Is there any characterized compound in paprika extract able to reach EPR after systemic administration?
- How do the authors explain the differences among the results obtained with FYP an FOP?
Author Response
Response to the Reviewer’s Comments (Reviewer 2)
Manuscript number: Nutrients-1003076
Title: Protective Effects of Fermented Paprika Extract (Capsicum annuum L.) on Sodium Iodate-induced Retinal Damage
We appreciate very much reviewers’ kind and helpful assessment of our manuscript. We also thank the reviewer for the effort and time put into the review of the manuscript. We have carefully considered and addressed reviewers’ comments, point by point. The changes in the revised manuscript have been marked with colored letters.
Responses on the specific comments and changes in the revised manuscript are as follows:
In their article the authors found that antioxidant properties of fermented paprika may be interesting to prevent oxidative stress cellular damage in RPE. Although the results are interesting there are some methodological concerns that require clarification.
Minor points:
- The meaning of the abbreviations FYP and PFO should appear in the main text. I only could find those meanings in a table legend.
In response to your advice, we have modified the abbreviations entirely to clarify the meaning and reduce confusion.
- Page 2 lines 54 to 60. The introduction should include the objectives and the antecedents of the research, I suggest to revise the redaction of those paragraphs.
Page 2 lines 54 to 60. As you mentioned, this paragraph has been revised to include the research goals and prior reports.
- Page 3 line 109. HEK293 cells were not used in any of the experiments.
We are so sorry for having a mistypo. It was corrected.
- Page 9 line 301. The authors should revise the sentence since they did not measure vision
Page 9, line 301. Following your advice, we have modified the sentence to fit the content of the study.
Major points:
- Statistical analysis. There are six groups. They should be compared together and not in groups of two or three
As you pointed out, we compared 6 groups rather than 2 or 3 groups, the results were similar to the previous one, and the material and method statistical analysis was revised.
- Although there are six animal per group, the results are expressed as the mean of three separate experiments. Is that enough for an ex vivo treatment?
We are so sorry for having a mistypo. It was corrected.
- Western blot results: How was calculated the “relative protein level” shown in bar graphs? Especially in those from those obtained from tissues from animals under the different treatments. It seemed to me that they quantified the fold increase comparing to “N” group”. Is the result the same when comparing the normalized densitometry of each group? There are some differences between the bars and the representative blots that should be carefully revised.
For statistical analysis of western blot bands, each band is firstly quantified through image analysis, and then normalized against the quantitative result of corresponding to each total band or value of beta actin. Secondly, the normalized value was expressed as a relative fold change against “N” group.
- Is there any characterized compound in paprika extract able to reach EPR after systemic administration?Very unfortunately this may not be the exact answer to the question, we were unable to find any significant changes at the level of single compound in the paprika extract after fermentation.
Very unfortunately this may not be the exact answer to the question, we were unable to find any significant changes at the level of single compound in the paprika extract after fermentation.
- How do the authors explain the differences among the results obtained with FYP an FOP?
We think it is difficult at this time to absolutely compare the difference between the results of FYP and FOP. The same is in raw materials, so further research may explain it.
Best Regards,
Submission Date
31 Oct 2020
Date of this review
30 Nov 2020

Reviewer 3 Report
Dear Authors,
The manuscripts needs major revisions
kind regards
This work is an interesting study related to the natural alternative treatment for retinal damage. The work is based on the study of the specific papikra extract and its activity in vitro and in vivo in mice.
The approach is interesting, however, many aspects should be addressed by authors.
- It is not clear (in this manuscript, in introduction section) the antecedents of previous studies with papikra or if this is the first work, and if so, specify it.
- In Material & Methods (line 95-97), It is not specified if the extract without fermerting is extracted with some type of liquid ... I suspect that it is only crushing. however, only the crushing generates an extract that can then be examined for its antioxidant properties as well as the tests on cells and mice ... it is not very clear...how are those “extracts”
- In Material & Methods (line 101), in section 2.3. Biochemical Analysis, what is meant by biochemical analysis…in general, biochemical analysis is aimed at blood tests… Is it a blood test or is it an extract test?
- In Material & Methods (line 137), in section 2.8. Animals and Treatment, the number of mice is not specified... what does SI mean? in no previous section does it speak of working with animals, it should be indicated
- In Results (line 187), data not shown corresponds to a publication or works that are not included in this study…clarify
- In Results (line 195), the increase is statistically significant???
- In Table 1. the description of the table generates confusion, the acronyms are wrong ... check line 202-203
- In Results (line 211-217), no cytotoxicity is observed but an effect on cell viability can explain it.
- In Figure 1a and 1b, (line 211-217), the cell viability of the YP extract is higher than FYP, and in Figure 1b the cell viability of the OP extract is higher than FOP, however, in text (line 213-214) it is different…please clarify
- In Line 262, clarify if pretreatment is done
- The discussion is a presentation of the results again, it is not compared with other studies ... much improvement is needed
Minor comments;
-Please actualize the references.
-I understand that authors are not native English speakers, however, grammar and typo errors strongly difficult the understanding of the work.
-Discussion section must be improved.
Author Response
Response to the Reviewer’s Comments (Reviewer 3)
Manuscript number: Nutrients-1003076
Title: Protective Effects of Fermented Paprika Extract (Capsicum annuum L.) on Sodium Iodate-induced Retinal Damage
We appreciate very much reviewers’ kind and helpful assessment of our manuscript. We also thank the reviewer for the effort and time put into the review of the manuscript. We have carefully considered and addressed reviewers’ comments, point by point. The changes in the revised manuscript have been marked with colored letters.
Responses on the specific comments and changes in the revised manuscript are as follows:
The manuscripts needs major revisions
kind regards
This work is an interesting study related to the natural alternative treatment for retinal damage. The work is based on the study of the specific papikra extract and its activity in vitro and in vivo in mice.
The approach is interesting, however, many aspects should be addressed by authors.
- It is not clear (in this manuscript, in introduction section) the antecedents of previous studies with papikra or if this is the first work, and if so, specify it.
Following your advice, we have revised the introduction. Revision contents are marked with a red color.
- In Material & Methods (line 95-97), It is not specified if the extract without fermerting is extracted with some type of liquid ... I suspect that it is only crushing. however, only the crushing generates an extract that can then be examined for its antioxidant properties as well as the tests on cells and mice ... it is not very clear...how are those “extracts”
In Material & Methods (line 95-97 in original article) corrected the preparation process of fermented and raw paprika powder and its extract. In this study, we lyophilized powder before and after fermentation, used it for animal testing, and the lyophilized powder was extracted with 70% ethanol, used for in vitro study.
- In Material & Methods (line 101), in section 2.3. Biochemical Analysis, what is meant by biochemical analysis…in general, biochemical analysis is aimed at blood tests… Is it a blood test or is it an extract test?
In Materials and methods (line 101 in original manuscript), biochemical analysis were modified to physicochemical analysis.
- In Material & Methods (line 137), in section 2.8. Animals and Treatment, the number of mice is not specified... what does SI mean? in no previous section does it speak of working with animals, it should be indicated
In Materials and Methods (line 137 in original manuscript), the method for animal experiments, the number of mice, and the meaning of SI were modified.
- In Results (line 187), data not shown corresponds to a publication or works that are not included in this study…clarify
Through our own prior research, we analyzed the vitamin C content of raw paprika (red, purple, yellow, and orange) and confirmed that it was high in orange- and yellow-paprika. Based on vitamin C content data (Figure. 1), we further analyzed using orange- and yellow-paprika in an ultraviolet-induced cataract animal model. These results were not reported under confidentiality agreements, but we have conducted further research based on these results. The result section has been modified.
Figure 1. Comparison results of vitamin C content (VC) in various colored-paprika (P). R, red; O, orange; Y, yellow; P, purple.
Figure 1. Effect of P (-O or -Y) on UVB-induced cataracts and oxidative stress. A. P (-O or –Y, 100 mg/kg) was administered orally once a day for one week. After 1 week UVB (120 mJ) was exposed to mice for 6 hours. Representative eye images have been presented. B. SOD activity and C. GSH levels were measured in each mouse eye tissue. *P < 0.05 vs. N (normal), #P < 0.05, ####P < 0.001 vs. C (control, UVB-exposed control). Values are means ± SEM. Significant differences between the groups were determines by ANOVA followed by Duncan’s multiple range test.
- In Results (line 195), the increase is statistically significant???
The results was not statistically signficant (or data change)
As you pointed out, the results in line 195 of the original manuscript were not statistically significant. However, the contents in both types of paprika showed the increasing pattern. The description in results section was corrected.
Furthermore, there was an error in the significance analysis of the vitamin C content and antioxidant effect.
- In Table 1. the description of the table generates confusion, the acronyms are wrong ... check line 202-203
Following your advice, we have modified the legend and acronyms in Table 1.
- In Results (line 211-217), no cytotoxicity is observed but an effect on cell viability can explain it.
The confusing results shown are due to errors in Figure 1A and B, the symbol of raw and fermented paprika was changed. As results was also corrected. We are so sorry for the confusion in the interpretation. Figure 1 and results have been corrected.
- In Figure 1a and 1b, (line 211-217), the cell viability of the YP extract is higher than FYP, and in Figure 1b the cell viability of the OP extract is higher than FOP, however, in text (line 213-214) it is different…please clarify
In the same line above response, the confusing results shown are due to errors in Figure 1A and B, the symbol of raw and fermented paprika was changed. As results was also corrected. We are so sorry for the confusion in the interpretation. Figure 1 and results have been corrected.
In Line 262, clarify if pretreatment is done
Following your advice, the sentence (line 262 in original manuscript) was change.
- The discussion is a presentation of the results again, it is not compared with other studies ... much improvement is needed
Following your comments, we thoroughly corrected and rewrote the manuscript including abstract, introduction, materials and method, results, and discussion.
Minor comments;
-Please actualize the references.
We confirmed the references.
-I understand that authors are not native English speakers, however, grammar and typo errors strongly difficult the understanding of the work.
We corrected the manuscript entirely including typos and grammar.
-Discussion section must be improved.
As you mentioned, we improved the discussion section.
Best Regards,
Submission Date
31 Oct 2020
Date of this review
30 Nov 2020

Round 2
Reviewer 1 Report
There are some concerns remains:
1) The N value for each group in all the experiments still missing. The authors must add it for scientific rigor.
2) In the reply, the authors did not address why they did not keep the tests/experiments in one cell model. What is the limitation of each model?
Author Response
Response to the Reviewer’s Comments
Manuscript number: Nutrients-1003076R1
Title: Protective Effects of Fermented Paprika Extract (Capsicum annuum L.) on Sodium Iodate-induced Retinal Damage
We are very grateful for the generous evaluation of repeated errors in previous review manuscripts.
We also thank the reviewers for the effort and time put into reviewing the manuscript. The changes in the revised manuscript have been marked with colored letters.
Responses on the specific comments and changes in the revised manuscript are as follows:
1) The N value for each group in all the experiments still missing. The authors must add it for scientific rigor.
As you pointed out, we rigorously reviewed and added the missing N values for each group in all experiments.
2) In the reply, the authors did not address why they did not keep the tests/experiments in one cell model. What is the limitation of each model?
The reason for confirming the effect of fermented paprika using various cell lines is that the most important purpose of this study was to focus on the improvement of the oxidative stress of paprika after fermentation and the effect on the pathogenesis caused by oxidative stress.
In Table 1, a significant difference was found in the radical scavenging activity of paprika after fermentation, but we tried to clarify the effect of fermentation by comparing it through various cell lines.
Best Regards,
Submission Date
31 Oct 2020
Date of this 2nd review
07 Dec 2020

Reviewer 3 Report
Minors revisions
Author Response
Response to the Reviewer’s Comments
Manuscript number: Nutrients-1003076R1
Title: Protective Effects of Fermented Paprika Extract (Capsicum annuum L.) on Sodium Iodate-induced Retinal Damage
We are very grateful for the generous evaluation of previous review manuscripts.The changes in the revised manuscript have been marked with colored letters.
Best Regards,
Submission Date
31 Oct 2020
Date of this 2nd review
07 Dec 2020